# Triple combination therapy of favipiravir plus two monoclonal antibodies eradicates influenza virus from nude mice

Maki Kiso[1], Seiya Yamayoshi ✉[1] & Yoshihiro Kawaoka [1,2,3]✉

Prolonged treatment of immunocompromised influenza patients with viral neuraminidase (NA) inhibitors is required, because the immune system of such patients fails to eradicate the viruses. Here, we attempted to eradicate influenza virus from the respiratory organs of nude mice, which is a model of immunocompromised hosts, by using combination therapy of the viral polymerase inhibitor favipiravir and monoclonal antibodies (mAbs) against the receptor-binding site (RBS) and stem of viral hemagglutinin (HA). Although monotherapy or combination therapy of two antivirals (two mAbs or favipiravir plus a mAb) suppressed virus replication, they failed to eradicate viruses from nude mice. In contrast, the triple combination therapy of favipiravir plus anti-Stem and anti-RBS mAbs completely stopped virus replication in nude mice, resulting in virus clearance. Triple combination approaches should be considered for the treatment of human immunocompromised patients with severe influenza.

[1] Division of Virology, Department of Microbiology and Immunology, Institute of Medical Science, University of Tokyo, Tokyo, Japan. [2] Department of Pathobiological Sciences, School of Veterinary Medicine, University of Wisconsin-Madison, Madison, Wisconsin, USA. [3] Department of Special Pathogens, International Research Center for Infectious Diseases, Institute of Medical Science, University of Tokyo, Tokyo, Japan. ✉email: yamayo@ims.u-tokyo.ac.jp; yoshihiro.kawaoka@wisc.edu

Treatment of seasonal influenza has relied on neuraminidase (NA) inhibitors and a polymerase inhibitor, which target the viral sialidase and polymerase activities of the virus, respectively. The time to influenza symptom alleviation is similar between one of the polymerase inhibitors, baloxavir marboxil, and the NA inhibitors; however, treatment with baloxavir marboxil efficiently reduces the viral load compared with treatment with an NA inhibitor within 1 day of initiation of treatment[1]. Another polymerase inhibitor, favipiravir (FAV), which is also known as T705, was approved in Japan in 2014 with an indication limited to treatment of novel or re-emerging influenza virus infections unresponsive or insufficiently responsive to available antivirals[2]. Mutant viruses resistant to FAV are rarely isolated in vitro and in vivo with one exceptional case[3]. Although detection of viruses that exhibit reduced susceptibility to NA inhibitors or baloxavir marboxil in immunocompetent patients has frequently been reported[1,4–9], such viruses usually do not dominate susceptible viruses with the exception of the worldwide spread of oseltamivir-resistant H1N1 virus in the 2007–2008 season. Recently, broadly protective human monoclonal antibodies (mAbs) against conserved regions of HA, including the receptor-binding site (RBS) and stem region, have been evaluated[10–16] and studies for their clinical application are being conducted[17–20]. However, no human mAbs are currently available for clinical use.

Virus clearance is usually delayed in immunocompromised patients, because the immune responses of such patients are insufficient to suppress virus replication. To inhibit virus replication in these patients, NA inhibitors are widely used but they fail to eradicate the virus from the respiratory organs[21–23]. Therefore, prolonged treatment with NA inhibitors is required for immunocompromised patients, which leads to frequent emergence of viruses that are resistant to these drugs[21–23]. The resistant viruses, in turn, compromise the effectiveness of the NA inhibitors, resulting in substantial mortality among immunocompromised patients[24]. In contrast, polymerase inhibitors have not been well studied as a treatment for immunocompromised patients. A reliable approach to achieve rapid eradication of the influenza virus from immunocompromised patients is required.

As an immunocompromised patient model for influenza infection, the nude mouse, which is immunologically deficient, because it lacks a thymus, has been used since 1981[25]. In nude mice infected with influenza virus, virus clearance is delayed and the survival rate is reduced compared with wild-type mice[26]. In our previous study, monotherapy with an NA inhibitor (oseltamivir or laninamivir) for 28 days did not improve the survival time of infected nude mice, whereas monotherapy with a virus polymerase inhibitor (FAV or baloxavir) or combination therapy of FAV plus an NA inhibitor (oseltamivir or laninamivir) for 28 days increased it[26,27]. Although these treatments were effective in terms of survival, they fail to eradicate the virus from the respiratory organs of the infected nude mice, resulting in relapse and death due to the remaining infectious viruses after termination of treatment. The development of a protocol to eradicate influenza virus from nude mice could contribute to improved treatment outcomes in human immunocompromised patients. Here we attempted to eradicate influenza virus from nude mice by using combination therapy of FAV and mAbs against the RBS and stem of HA.

## Results

### Survival of nude mice that received each antiviral therapy. To assess virus clearance from an immunocompromised host, we intranasally infected nude mice with $10^3$ plaque-forming units (PFUs) of MA-CA04. The infected mice were treated with FAV at 100 mg/kg once a day for 28 days, a human mAb against the HA stem (clone CR9114) at 5 mg/kg once per 3 days for 14 days, or a human mAb against the RBS of HA (clone F3A19) at 1 mg/kg once per 3 days for 14 days, alone or in combination (Table 1). Methyl cellulose plus phosphate-buffered saline (PBS) and a human mAb against the HA of influenza B virus were administrated as negative controls. Body weight change and survival of these mice were monitored for 188 days. All of mice that received methyl cellulose plus PBS or the anti-B HA mAb died within 8 days of infection; median survival was 6 days (Fig. 1 and Table 1). Infected mice that were treated with FAV alone died between 38 and 42 days post infection (median survival, 40 days), whereas two or three out of four mice that received the anti-Stem or anti-RBS mAb alone died within 8 days post infection, and the remaining mice died at 51 or 37 days post infection (median survival, 29.5 or 8 days). Monotherapy with FAV or the anti-RBS mAb significantly increased the survival time of the infected mice compared with the methyl cellulose plus PBS-treated group ($P <$ 0.01; log-rank test followed by Benjamini–Hochberg correction). Mice treated with the combination of FAV plus anti-Stem mAb, FAV plus anti-RBS mAb, or anti-Stem mAb plus anti-RBS mAb survived for significantly longer than the control groups, but died between 39 and 155 days post infection (median survival, 119.5, 71.5, or 45 days, $P < 0.01$). The hazard ratio was decreased by all treatments tested (Table 1). In all lungs of dead mice after termination of the combination therapies with two antivirals, viruses were detected at 5.7–6.9 $\log_{10}$ PFU/g, indicating that the viruses were not eradicated from the nude mice by the combination therapies with two antivirals. In contrast, all four mice that received the triple combination therapy of FAV, anti-Stem mAb, and anti-RBS mAb survived for 188 days (Fig. 1, $P < 0.01$) and did not harbor detectable viruses in the lung at day 188. These results show that the triple combination therapy of FAV, anti-Stem mAb, and anti-RBS mAb can eradicate influenza virus from immunocompromised hosts.

### Virus titers in nude mice that received each treatment. To evaluate the degree of virus clearance, three nude mice per group infected with $10^3$ PFU of MA-CA04 were treated as described in Table 1 and virus titers were measured in the lungs of treated mice at 7, 14, and 28 days post infection. The virus titers in the lungs of the mice that received methyl cellulose plus PBS or anti-B HA mAb were 7.4 or 7.2 $\log_{10}$ PFU/g, respectively, at day 7 (Table 2). Compared with the methyl cellulose plus PBS-treated group, the virus titers in the lungs of mice at day 7 were significantly decreased by treatment with FAV, FAV plus anti-Stem mAb, FAV plus anti-RBS mAb, or FAV plus both mAbs, but were not affected by treatment with anti-Stem mAb, anti-RBS mAb, or anti-Stem mAb plus anti-RBS mAb (Table 2). Virus titers at day 14 post infection were significantly reduced by treatment with FAV plus anti-Stem mAb, FAV plus anti-RBS mAb, anti-Stem mAb plus anti-RBS mAb, or FAV plus both mAbs compared with the anti-Stem mAb-treated group. At day 28 post infection, virus titers were significantly reduced by treatment with FAV, FAV plus anti-Stem mAb, FAV plus anti-RBS mAb, or FAV plus both mAbs compared with the anti-Stem mAb-treated group (Table 2). Of note, no infectious viruses or viral RNA were detected at day 28 in the lungs of mice treated with FAV plus Anti-RBS mAb or FAV plus both mAbs by plaque assay and reverse-transcriptase PCR (RT-PCR) targeting to the NS segment. Taken together with the survival data, these results demonstrate that the triple combination therapy of FAV, anti-Stem mAb, and anti-RBS mAb can eradicate virus from the lungs of nude mice. Viruses and viral genomes were not detected in the lungs of mice treated with FAV plus anti-RBS mAb, but these mice died between 51 and 100 days post infection. The cause of death of these mice remains

**Table 1 Summary of treatment groups.**

| Group number | Treatment with | Concentration(s) | Treatment regimen | Route of administration | Number of mice[b] | Median survival days | Hazard ratio | Median |
|---|---|---|---|---|---|---|---|---|
| 1 | Methyl cellulose + PBS | 0.5% + 1× | Daily for 28 days + once per 3 days for 14 days | Oral and intraperitoneal | 7 | 6 | 1 | |
| 2 | Anti-B HA mAb | 15 mg/kg | Once per 3 days for 14 days | Intraperitoneal | 7 | 6 | 0.78 | |
| 3 | FAV[a] | 100 mg/kg | Daily for 28 days | Oral | 13 | 40 | 0.33 | |
| 4 | Anti-Stem mAb | 5 mg/kg | Once per 3 days for 14 days | Intraperitoneal | 13 | 29.5 | 0.53 | |
| 5 | Anti-RBS mAb | 1 mg/kg | Once per 3 days for 14 days | Intraperitoneal | 13 | 8 | 0.33 | |
| 6 | FAV + Anti-Stem mAb | 100 mg/kg + 5 mg/kg | Daily for 28 days + once per 3 days for 14 days | Oral + intraperitoneal | 13 | 119.5 | 0.33 | |
| 7 | FAV + Anti-RBS mAb | 100 mg/kg + 1 mg/kg | Daily for 28 days + once per 3 days for 14 days | Oral + intraperitoneal | 13 | 71.5 | 0.33 | |
| 8 | Anti-Stem mAb + Anti-RBS mAb | 5 mg/kg + 1 mg/kg | Once per 3 days for 14 days | Intraperitoneal + intraperitoneal | 13 | 45 | 0.33 | |
| 9 | FAV + Anti-stem mAb + Anti-RBS mAb | 100 mg/kg + 5 mg/kg + 1 mg/kg | Daily for 28 days + once per 3 days for 14 days + once per 3 days for 14 days | Oral + intraperitoneal + intraperitoneal | 13 | -[c] | - | |

[a]Favipiravir.
[b]Four mice for the survival study and 3 or 9 mice for the virus titer assessment (1 or 3 timepoints; 3 mice per timepoint).
[c]Undefined.

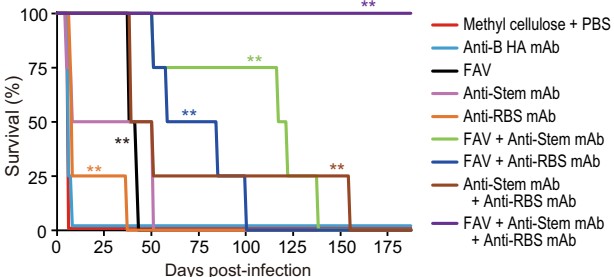

Fig. 1 Survival of nude mice infected with MA-CA04. Nude mice were infected with $10^3$ PFU of MA-CA04. Four infected mice per group were treated with FAV, anti-Stem mAb, or anti-RBS mAb alone or in combination ($n = 4$ biologically independent animals). Infected mice that received methyl cellulose plus PBS or anti-B HA mAb served as controls ($n = 4$ biologically independent animals). Statistically significant differences compared with methyl cellulose plus PBS were determined by use of the log-rank (Mantel–Cox) test followed by Benjamini–Hochberg correction. $^{**}P < 0.01$.

unknown; it is possible that virus was likely present in these mice, but was under the level of detection at day 28 and likely started to replicate upon discontinuation of the medication.

**Absence of reduced-sensitivity viruses upon treatment**. Emergence of drug-resistant mutants after long-term antiviral treatment is a major concern[28]. To examine whether such mutants emerged in nude mice after FAV treatment, we examined the sensitivity of viruses isolated from the lungs of killed and dead mice that were treated with FAV alone or in combination. The sensitivity of each isolate to FAV was measured by using plaque reduction assays. Based on the $IC_{50}$ values obtained, all tested viruses showed similar sensitivity to FAV as the wild-type virus (Table 3). As the viruses isolated from the mouse lungs might be a mixed population of wild-type virus and virus with reduced susceptibility to FAV, we purified three clones from the lungs of mice treated with FAV or FAV plus anti-Stem mAb and killed at 28 days post infection by plaque purification, and then tested the sensitivity of the plaque-purified viruses to FAV in a plaque reduction assay. The $IC_{50}$ values of all tested plaque-purified viruses to FAV were similar to that of wild-type virus, indicating that mutant viruses with reduced sensitivity to FAV did not emerge after treatment with FAV alone or in combination.

Emergence of mutant viruses that can escape from neutralizing mAbs after treatment with protective mAb is a major concern with mAb treatment[29]. To clarify whether such mutant viruses emerged after mAb treatment, we analyzed the genome sequence of viruses from the lungs of mice treated with anti-Stem or anti-RBS mAb alone or in combination. For this, we used the lung samples derived from mice killed at 14 days post infection, the day of treatment termination, for virus titration and from mice that died after 37 days post infection (Table 4). By Sanger sequencing, zero to five mutations were found in the HA of virus in the lung of mice treated with mAbs (Table 4). In particular, amino acid mutations in HA were detected in a higher proportion of viruses in the FAV plus anti-Stem mAb-treated mice than in the other groups tested. These amino acid mutations were mapped onto the three-dimensional structure of the H1–HA trimer. The amino acids at positions 125, 128, 186, 188, 192, and 198 mapped to the top of the HA head, the amino acids at positions 49, 390, and 392 mapped to the lower part of the HA head, and the amino acid at position 362 mapped to the HA stem (Fig. 2). We then asked whether these mutant viruses escaped from the anti-Stem and anti-RBS mAbs that we used for

**Table 2 Lung virus titers of infected mice treated with the indicated inhibitors.**

| Group number | Treatment with | Virus titer (mean log$_{10}$ PFU ± SD/g)[b] | | |
|---|---|---|---|---|
| | | Day 7 | Day 14 | Day 28 |
| 1 | Methyl cellulose + PBS | 7.4 ± 0.1 | ND[d] | ND |
| 2 | Anti-B HA mAb | 7.2 ± 0.3 | ND | ND |
| 3 | FAV[a] | 6.1 ± 0.2[*] | 5.9 ± 0.3 | 5.5 ± 0.3[**] |
| 4 | Anti-Stem mAb | 7.1 ± 0.2 | 7.0 ± 0.4 | 7.1 ± 0.8 |
| 5 | Anti-RBS mAb | 6.3 ± 0.8 | 5.2 ± 1.8 | 7.0, 6.9, NA[e] |
| 6 | FAV + Anti-Stem mAb | 4.2 ± 0.4[**] | 3.9 ± 0.5[**] | 4.6 ± 0.8[**] |
| 7 | FAV + Anti-RBS mAb | 2.9, <1.7[c], 3.4[**] | <1.7, <1.7, 4.1[**] | <1.7, <1.7, <1.7[**] |
| 8 | Anti-Stem mAb + Anti-RBS mAb | 6.6 ± 0.6 | 3.7 ± 0.7[**] | 6.5 ± 0.7 |
| 9 | FAV + Anti-Stem mAb + Anti-RBS mAb | <1.7, 2.0, <1.7[**] | <1.7, <1.7, <1.7[**] | <1.7, <1.7, <1.7[**] |

Statistically significant differences compared with group 1 (day 7) or 4 (days 14 and 27) were determined by use of a one-way analysis of variance followed by a Dunnett test.
[*]$P < 0.05$ and [**]$P < 0.01$, respectively.
[a]Favipiravir.
[b]BALB/c-nu/nu mice were intranasally inoculated with 103 PFU of MA-CA04 virus. Three animals per group were euthanized on days 7, 14, and 28 post infection.
[c]Detection limit is 1.7 log10 PFU/g.
[d]Not done.
[e]Not available, because mouse succumbed to infection before the day of sampling.

treatment. The single mutation of D188N, which was detected in the HA of virus in mice treated with anti-RBS mAb, increased the IC$_{50}$ value to anti-RBS mAb (Table 5). The mutations of A49T, P125S, T198A, Q390H, and T392I increased the IC$_{50}$ value to anti-RBS mAb even though these mutations were detected in the HA of virus found in mice treated with FAV and anti-Stem mAbs (Table 5). However, the level of reduced sensitivity to the anti-RBS mAb was minimal. The IC$_{50}$ values to anti-Stem mAb were not affected by any mutation tested (Table 5). These data indicate that mutant viruses that can escape from mAbs rarely appear in nude mice after long-term mAb treatment.

## Discussion

Antiviral treatment is widely used in immunocompromised patients to inhibit influenza virus replication, because such patients' immune system fails to suppress virus replication. In our previous nude mouse model study, mono- or combination therapy of NA inhibitors (oseltamivir or laninamivir) and/or virus polymerase inhibitors (FAV or baloxavir) increased survival time but failed to achieve virus clearance, resulting in death after termination of treatment[26,27]. In the present study, we attempted to eradicate viruses from the respiratory organs of nude mice by using combination therapy of FAV and two mAbs against the HA RBS and stem. Although treatment with FAV or a combination of two antivirals improved the survival time of infected nude mice, all of the mice that received such treatments died with high virus titers in their lungs during the observation period, suggesting that these treatments failed to achieve virus clearance from the respiratory organs. In contrast, influenza virus was successfully eradicated from nude mice by the triple combination therapy of FAV and mAbs against the HA RBS and stem: all mice that received this treatment survived for 188 days and did not harbor infectious viruses in the lungs at 188 days post infection. Our data thus suggest that this triple combination therapy can completely stop virus replication. Triple combination therapy of antivirals with different mechanisms of action might cause a strong synergistic effect, which would support a triple combination approach to the treatment of human immunocompromised patients with influenza.

For resistant viruses to emerge, they must replicate under the suppressive pressure of antivirals. Therefore, it is important to sufficiently suppress or ultimately block virus replication to

**Table 3 Susceptibility of isolated viruses to FAV.**

| Group number | Treatment with | Days post infection | IC$_{50}$ value[a] (µg/ml) |
|---|---|---|---|
| 3 | FAV | **28**[b] | **2.3** |
| | | **28** | **2.1** |
| | | **28** | **1.6** |
| | | 38 | 2.0 |
| | | 38 | 1.7 |
| | | 42 | 1.7 |
| | | 43 | 2.3 |
| 6 | FAV + Anti-Stem mAb | **28** | **1.8** |
| | | **28** | **1.7** |
| | | **28** | **2.0** |
| | | 51 | 1.9 |
| | | 117 | 1.8 |
| | | 122 | 2.3 |
| | | 138 | 2.3 |
| 7 | FAV + Anti-RBS mAb | 28 | NA[c] |
| | | 28 | NA |
| | | 28 | NA |
| | | 51 | 1.1 |
| | | 58 | 4.7 |
| | | 85 | 1.1 |
| | | 100 | 2.3 |

[a]IC$_{50}$ value of wild-type virus to FAV was 1.3 µg/ml.
[b]Bolded numbers indicated that three out of three plaque-purified viruses were susceptible to FAV.
[c]Virus was not isolated.

prevent the emergence of resistant viruses. Our data show that the triple combination therapy of FAV, anti-Stem mAb, and anti-RBS mAb completely stopped virus replication in infected nude mice, leading to virus clearance. Our triple combination therapy is, therefore, less likely than current monotherapies to lead to the emergence of resistant virus in immunocompromised hosts.

Even though the combination therapy of FAV plus the anti-RBS mAb reduced virus titers in the lung to below the detection limit during drug treatment, infectious viruses have likely remained at the termination of treatment, leading to the deaths of nude mice likely due to virus replication. This finding indicates that termination of treatment when the virus titer in the tracheal

**Table 4 Amino acid substitutions in HA of viruses isolated from lungs of treated mice.**

| Group number | Treatment with | Days post infection | Amino acid mutation(s) in HA[a] |
|---|---|---|---|
| 4 | Anti-Stem mAb | 14 | None |
| | | 14 | None |
| | | 14 | None |
| | | 51 | None |
| | | 51 | L192I |
| 5 | Anti-RBS mAb | 14 | D188N |
| | | 14 | None |
| | | 14 | None |
| | | 37 | None |
| 6 | FAV + Anti-Stem mAb | 14 | None |
| | | 14 | V200I and S327Y |
| | | 14 | None |
| | | 51 | D128E |
| | | 117 | A49T, P125S, T198A, Q390H, and T392I |
| | | 122 | L192I |
| | | 138 | L192I, T509A, and R516W |
| 7 | FAV + Anti-RBS mAb | 14 | NA[b] |
| | | 14 | NA |
| | | 14 | None |
| | | 51 | None |
| | | 58 | None |
| | | 85 | None |
| | | 100 | L192I |
| 8 | Anti-Stem mAb + Anti-RBS mAb | 14 | None |
| | | 14 | None |
| | | 14 | None |
| | | 39 | None |
| | | 39 | None |
| | | 51 | None |
| | | 155 | S186N, L192I, Y362H, and R516G |

[a]H1 numbering.
[b]Virus was not isolated.

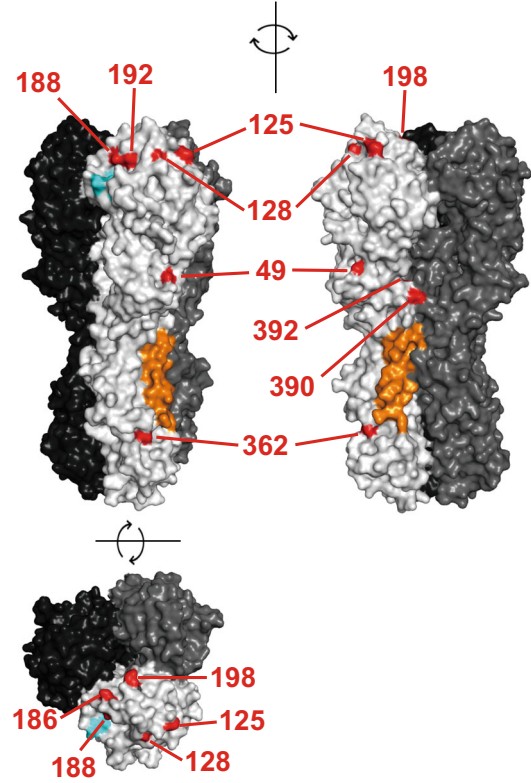

**Fig. 2 Mapping of mutations found in viruses isolated from mAb-treated mice.** Amino acid substitutions found in the HA of viruses isolated from the lungs of mice treated with anti-Stem and anti-RBS mAbs were mapped onto the 3D structure of the H1–HA trimer (PDB; 3LZG) by using the molecular graphics system PyMOL. The amino acid at position 200 is not exposed on the molecular surface of this model and the amino acids at positions 327, 509, and 516 are not included in this model. Cyan indicates amino acids involved in receptor binding and orange indicates alpha helix A in the HA stem, which is the major epitope in the HA stem. Mutations shown in red are shown with H1 numbering.

swab sample of an immunocompromised patient is below the detection limit carries a risk of relapse due to remaining infectious viruses. Therefore, we contend that immunocompromised influenza patients should be closely monitored after the termination of treatment.

## Methods

**Ethics**. All animal experiments were conducted in accordance with the University of Tokyo's Regulations for Animal Care and Use, which were approved by the Animal Experiment Committee of the Institute of Medical Science, the University of Tokyo. The committee acknowledged and accepted both the legal and ethical responsibility for the animals, as specified in the Fundamental Guidelines for Proper Conduct of Animal Experiment and Related Activities in Academic Research Institutions under the jurisdiction of the Ministry of Education, Culture, Sports, Science, and Technology of Japan.

**Mouse infection test**. For the mouse survival test, four 6-week-old female nude mice (BALB/c-*nu/nu*; Japan SLC, Inc.) per group were anesthetized with

**Table 5 Sensitivity of HA mutant viruses to mAbs.**

| Amino acid mutation(s) in HA[a] | IC$_{50}$ values against | |
|---|---|---|
| | Anti-RBS mAb | Anti-Stem mAb |
| Wild-type | 0.19 | 5.0 |
| D128E | 0.46 | 9.9 |
| D188N | **4.4**[b] | 25 |
| L192I | 0.20 | 4.4 |
| V200I and S327Y | 0.17 | 2.5 |
| L192I, T509A, and R516W | 0.46 | 8.8 |
| S186N, L192I, Y362H, and R516G | 0.78 | 6.3 |
| A49T, P125S, T198A, Q390H, and T392I | **2.5** | 5.0 |

[a]H1 numbering.
[b]Bolded numbers indicate reduced sensitivity to the mAb. An eight times higher IC$_{50}$ value was considered to reflect reduced sensitivity.

isoflurane and intranasally infected with $10^3$ PFUs of mouse-adapted A/California/04/2009 (H1N1pdm09; MA-CA04)[30]. The infected mice were treated with FAV (100 mg/kg), the anti-Stem mAb clone CR9114[11] (5 mg/kg), and the anti-RBS mAb clone F3A19[31] (1 mg/kg) alone or in combination (Table 1). Treatment was administered orally (FAV) or intraperitoneally (anti-Stem and Anti-RBS

mAbs) once daily for 28 days (FAV) or once every 3 days for 14 days (anti-Stem and anti-RBS mAbs). Methyl cellulose (0.5%) and PBS served as controls for the oral and intraperitoneal treatments, respectively. All treatments were initiated at 1 h post infection. Survival and clinical signs were monitored daily for 188 days. Changes in body weight and survival were monitored daily for 30 days or 5–6 days per week after 31 days post infection. Mice that lost 25% or more of their initial body weight were scored as dead and killed in accordance with institutional guidelines.

For assessment of virus titers in mouse lungs, three or nine 6-week-old female nude mice per group were infected and treated as described above. On days 7, 14, and 28 post infection, three randomly selected mice per group were killed and virus titers in the lungs were determined by using plaque assays in Madin-Darby canine kidney (MDCK) cells. Viral RNA was extracted from lung samples collected at 28 days post infection and RT-PCR was performed using primers against the NS segment.

**Sensitivity to FAV**. Using the lungs of mice that received treatments that included FAV, we attempted to isolate viruses using MDCK cells in 24-well plates. The isolated viruses were titrated and ~50 PFU of isolated viruses were then inoculated into confluent MDCK cells in 6-well plates. After infection, the cells were overlaid with MEM containing 0.3% bovine serum albumin, 1% agarose, 1 μg/ml TPCK (N-tosyl-L-phenylalanine chloromethyl ketone)-treated trypsin, and various concentrations (0.01–100 nM) of FAV. The plates were the incubated at 37 °C for 2–3 days and plaques were counted to determine $IC_{50}$ values.

**Sequence analysis of HA**. Viral RNA was extracted from the supernatants of virus-infected MDCK cells by using the QIAamp Viral RNA Mini Kit (Qiagen). The cDNA was synthesized by using Superscript III reverse transcriptase (Invitrogen) and a U12 primer (5′-AGCAAAAGCAGG-3′). The cDNA products were amplified by PCR using primers specific for the HA segment. The PCR products were sequenced with the BigDye terminator 3.1 kit on an ABI 3130xl (Applied biosystems). The sequence data were submitted to Genbank (accession numbers LC537233–LC537240).

**Virus neutralization test**. Two-fold serially diluted purified anti-Stem or anti-RBS mAb was mixed with 100 $TCID_{50}$ of the isolated virus and then incubated at 37 °C for 30 min. The mixtures were inoculated into MDCK cells in quadruplicate and incubated for 1 h at 37 °C. BSA-MEM containing TPCK-treated trypsin was added to each well and the cells were incubated for 3 days at 37 °C. The cytopathic effect was examined and antibody titers required to reduce virus replication by 50% ($IC_{50}$) were determined by using the Spearman–Karber formula.

**Structural analysis**. Amino acid positions were plotted on the crystal structure of CA04 HA (PDB accession code, 3LZG) by using the PyMOL molecular graphics system to visualize the trimer.

**Statistics and reproducibility**. One-way analysis of variance followed by Dunnett's test and the log-rank (Mantel–Cox) test followed by Benjamini–Hochberg correction were performed by using GraphPad Prism 7.04. Each sample size is defined in the main text, footnotes, or legends. No samples were excluded from the analysis.

**Reporting summary**. Further information on research design is available in the Nature Research Reporting Summary linked to this article.

## Data availability

All data analyzed during this study are included in this article. The sequence data were submitted to Genbank (accession numbers LC537233–LC537240). The datasets generated and analyzed during the current study are available from the corresponding author on reasonable request.

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

## Acknowledgements
We thank Dr Susan Watson for editing the manuscript. This work was supported by Leading Advanced Projects for medical innovation (LEAP) (JP19am001007), the Japan Initiative for Global Research Network on Infectious Diseases (J-GRID) (JP19fm0108006), and a Research Program on Emerging and Re-emerging Infectious Diseases (JP19fk0108104) from the Japan Agency for Medical Research and Development (AMED), by Grants-in-Aid for Scientific Research on Innovative Areas from the Ministry of Education, Culture, Science, Sports, and Technology (MEXT) of Japan (numbers 16H06429, 16K21723, and 16H06434), by JSPS KAKENHI Grant Number 18KO7140, and by the Center for Research on Influenza Pathogenesis (CRIP) funded by NIAID Contract HHSN272201400008C.

## Author contributions
M.K., S.Y., and Y.K. designed this study. M.K. performed the experiments. S.Y. and Y.K. analyzed the data and wrote the manuscript. All of the authors reviewed the manuscript and approved the final version.

## Competing interests
Y.K. has received speaker's honoraria from Toyama Chemical and Astellas, Inc., has received grant support from Chugai Pharmaceuticals, Daiichi Sankyo Pharmaceutical, Toyama Chemical, Tauns Laboratories, Inc., Otsuka Pharmaceutical Co., Ltd, Denka Seiken Co., Ltd, and Shionogi & Co., Ltd, and is a co-founder of FluGen. The other authors have no conflicts of interest.
