## [Peer review file · Communications Biology]

Referee expertise:

Referee #1:

Referee #2:

Referee #3:

Reviewers' comments:

Reviewer #1 (Remarks to the Author):

1. Information on mean survival day and hazard ratio needs to be added in Table 1 as separate columns.
2. Since death of nude mice was still observed after administration of FAV + the anti-RBS mAb when the virus was not detected by plaque assay, absence of virus replication shown on Figure 1 and Table 2 needs to be confirmed by the second orthogonal method, for example, by PCR analysis. Specifically, the authors need to confirm virus absence after FAV+anti-RBS mAb and triple combination FAV+anti-stem mAb+anti-RBS mAb 28-day post-treatment.
3. The authors should indicate when exactly the treatment was initiated (for example, 4 hr after virus infection or immediately after virus infection).
4. It is not clear how isolated viruses were propagated for measuring FAV sensitivity. Were the viruses passaged 1 time in MDCK cells? It is not clear how much volume of the virus the authors were able to isolate from mice in order to be able to estimate 50 FPU for the following FAV sensitivity assessment. This needs to be clarified.

Reviewer #2 (Remarks to the Author):

The authors conducted an animal study to examine the in vivo effect of combination therapy of favipiravir and human mAbs to the receptor-binding site and stem of HA against H1N1 mouse adapted CA04 strain in immunocompromised mice. Several control groups were included in the experiments. The survival rates and lung virus titers were compared in different groups. The emergence of favipiravir-resistant and mAb-escape mutants were evaluated. The results showed that the triple combination therapy of favipiravir plus anti-stem and anti-receptor-binding site mAbs resulted in complete protection and virus clearance in nude mice.

I think this study contributes to the field of the therapeutic strategy for influenza virus infection in immunocompromised hosts by a descent study design and data analysis, although an optimal treatment protocol for control of severe influenza in such hosts remains debatable.

A few minor comments are as follows,

1. What is the rationale for the dose and frequency in the treatment regimen (Table 1)?
2. What is your best suggestion for the treatment protocol for severe influenza among immunocompromised hosts based on previous (refs 26, 27) and current study?
3. What is the role of Fc receptors in the mode of action of monoclonal antibodies for in vivo protection?

Reviewer #1

- 1. Information on mean survival day and hazard ratio needs to be added in Table 1 as separate columns.**

In response to the reviewer's comment, we have added the median survival day and hazard ratios (Page 5, lines 9–19 and Table 1).

- 2. Since death of nude mice was still observed after administration of FAV + the anti-RBS mAb when the virus was not detected by plaque assay, absence of virus replication shown on Figure 1 and Table 2 needs to be confirmed by the second orthogonal method, for example, by PCR analysis. Specifically, the authors need to confirm virus absence after FAV+anti-RBS mAb and triple combination FAV+anti-stem mAb+anti-RBS mAb 28-day post-treatment.**

In response to the reviewer's comment, we performed RT-PCR using 6 lung homogenate samples obtained from mice that were treated with FAV plus anti-RBS mAb or FAV + anti-stem mAb + anti-RBS for 28 days. Viral RNA was not detected by the RT-PCR. This result is now described in the main text (Page 6, lines 13–15 and Page 11, line 25–Page 12, line 2).

- 3. The authors should indicate when exactly the treatment was initiated (for example, 4 hr after virus infection or immediately after virus infection).**

In response to the reviewer's comment, we have added the start time of the treatment to the Methods section (Page 11, line 18).

- 4. It is not clear how isolated viruses were propagated for measuring FAV sensitivity. Were the viruses passaged 1 time in MDCK cells? It is not clear how much volume of the virus the authors were able to isolate from mice in order to be able to estimate 50 FPU for the following FAV sensitivity assessment. This needs to be clarified.**

In response to the reviewer's comment, we have provided additional information regarding how FAV sensitivity was measured (Page 12, lines 4–7).

Reviewer #2

- 1. What is the rationale for the dose and frequency in the treatment regimen (Table 1)?**

There have been no previous reports of testing double or triple combination therapies comprising favipiravir and monoclonal antibodies in a nude mouse infection model. Therefore,

we selected the treatment regimens on the basis of our previous in vivo protection studies with wild-type and nude mice (see refs. 10, 26, and 31).

2. *What is your best suggestion for the treatment protocol for severe influenza among immunocompromised hosts based on previous (refs 26, 27) and current study?*

As stated in the manuscript (Page 9, lines 16–18), we think that the triple combination therapy of favipiravir plus monoclonal antibodies against the HA RBS and stem is the best protocol so far.

3. *What is the role of Fc receptors in the mode of action of monoclonal antibodies for in vivo protection?*

We have not tested the role of Fc receptor-mediated antiviral activities in our nude mouse infection model. This point is important to understand the mechanism of virus eradication from hosts. Therefore, we will examine this in a future study.

Referee expertise:

Referee #1:

Referee #2:

Referee #3:

Reviewers' comments:

Reviewer #1 (Remarks to the Author):

The authors have responded on most of the questions. However, one finding still requires some clarification. The authors stated that "viruses and viral genomes were not detected in the lungs of mice treated with FAV plus anti-RBS mAb, but these mice died between 51-100 days p.i." (page 6, lines 17-19). The authors speculate that the virus replicated below the detection limit. It is very unlikely, because virus was not detected by two orthogonal methods (plaque assay and qPCR) at 28 days p.i. According to WHO protocols, if virus could not be detected by qPCR, this indicates absence of infection. The authors need to examine the real cause of death of mice treated with FAV plus anti-RBS mAb. It would be very beneficial for the manuscript, if the authors could provide the data on qPCR analysis between 28 days p.i. and mean survival day (71.5). The authors' speculation that "infectious viruses must have remained at the termination of treatment" (page 9 last line, page 10, first line) must be supported by real data.

Reviewer #2 (Remarks to the Author):

I am satisfied with the revised manuscript and have no further comments.

Reviewer #1

The authors have responded on most of the questions. However, one finding still requires some clarification. The authors stated that "viruses and viral genomes were not detected in the lungs of mice treated with FAV plus anti-RBS mAb, but these mice died between 51-100 days p.i." (page 6, lines 17-19). The authors speculate that that the virus replicated below the detection limit. It is very unlikely, because virus was not detected by two orthogonal methods (plaque assay and qPCR) at 28 days p.i. According to WHO protocols, if virus could not be detected by qPCR, this indicates absence of infection. The authors need to examine the real cause of death of mice treated with FAV plus anti-RBS mAb. It would be very beneficial for the manuscript, if the authors could provide the data on qPCR analysis between 28 days p.i. and mean survival day (71.5). The authors' speculation that "infectious viruses must have remained at the termination of treatment" (page 9 last line, page 10, first line) must be supported by real data.

Thank you for pointing out this discrepancy. In response, we have clarified our statement (Page 6, lines 19–21). What follows is a detailed explanation:

It is important to note that in our assay, we can only detect 1.7 log₁₀ PFU/g of virus in organs (Table 2); i.e., virus < 1.7 log₁₀ PFU/g cannot be detected by using our standard virologic method. Similarly, qPCR is not always sensitive especially for detecting virus genome in tissue homogenates or clinical samples; see Extended Data Fig. 1c of Imai et al. Nat Microbiol., 5, 27–33 (2020)(<https://www.nature.com/articles/s41564-019-0609-0>). Therefore, we respectfully disagree with the reviewer's point that "The authors speculate that that the virus replicated below the detection limit. It is very unlikely, because virus was not detected by two orthogonal methods (plaque assay and qPCR) at 28 days p.i".

Regarding the statement in the WHO protocols that "if virus could not be detected by qPCR, this indicates absence of infection), we believe that there are cases in which virus genome is not detected by qPCR, but animals are still infected.

It is also not clear what we would learn by repeating the experiments to find virus (genome) in animals treated with FAV plus anti-RBS mAb between days 28 p.i. and the mean survival day (71.5) because we already know that virus was detected in animals that died between 51 and 100 days p.i., but not in animals sacrificed at day 28 p.i.. The logical explanation is what we concluded: the virus (genome) in these animal was under the limit

of detection by the methods we used, but started to grow upon discontinuation of the medication and, eventually, the animals died.

Finally, and most importantly, regardless of whether or not virus (genome) is present in animals treated with FAV plus anti-RBS mAb between days 28 p.i. and the mean survival day (71.5), our conclusion that influenza virus can be eliminated from nude mice by the triple combination therapy of FAV plus two monoclonal antibodies still stands.